# Entanglement between Water Un-Extractable Arabinoxylan and Gliadin or Glutenins Induced a More Fragile and Soft Gluten Network Structure

**DOI:** 10.3390/foods12091800

**Published:** 2023-04-26

**Authors:** Fan Li, Tingting Li, Jiajia Zhao, Mingcong Fan, Haifeng Qian, Yan Li, Li Wang

**Affiliations:** 1State Key Laboratory of Food Science and Technology, School of Food Science and Technology, National Engineering Research Center for Functional Food, Jiangnan University, 1800 Lihu Avenue, Wuxi 214122, China; 6200112039@stu.jiangnan.edu.cn (F.L.);; 2Department of Food Science and Engineering, College of Light Industry and Food Engineering, Nanjing Forestry University, 159 Longpan Road, Nanjing 210037, China; 3College of Cooking Science and Technology, Jiangsu College of Tourism, Yangzhou 225000, China

**Keywords:** water-unextractable arabinoxylan, gliadins, glutenins, chemical interactions

## Abstract

This study aimed to investigate the effects of water-unextractable arabinoxylan (WUAX) on the gluten network structure, especially on gliadins and glutenins. The results indicated that the free sulfhydryl (free SH) of gliadins increased by 25.5% with 100 g/kg WUAX, whereas that of glutenins increased by 65.2%, which inhibited the formation of covalent bonds. Furthermore, β-sheets content decreased 5.63% and 4.75% for gliadins and glutenins with 100 g/kg WUAX, respectively, compared with the control. WUAX increased β-turns prevalence for gliadins, while the content of α-helixes and random coils had less fluctuation. In glutenins, the contents of α-helixes and β-sheets decreased and β-turns increased. Moreover, compared with the control, the weight loss rate for gliadins and glutenins increased by 2.49% and 2.04%, respectively, with 60 g/kg WUAX. The dynamic rheological analysis manifested that WUAX impaired the viscoelasticity property of gliadin and glutenin. Overall, WUAX weakened the structure of the gliadins and glutenins, leading to quality deterioration of gluten.

## 1. Introduction

Recently, many reports have shown that whole grains decreased the risk of chronic diseases, such as colorectal cancer, cardiovascular disease, and type 2 diabetes mellitus [1]. Thus, whole grain-based products are increasing rapidly for nutritional and health benefits. Compared with refined grains, whole grains have more healthy components, such as dietary fiber, micronutrients, minerals, and polyphenols [2], which are mainly located in the bran or germ. However, these components lead to a rough texture and a poor appearance, significantly affecting consumers’ desires. Wheat-based foods are the most widely consumed cereal products globally, among which whole wheat foods are increasingly popular for consumers [3]. Wheat bran, a by-product in the wheat flour processing process, is mainly composed of polysaccharides (56–66%), protein (15–22%), and lignin (4–8%) [4]. Arabinoxylan (AX) is an important component of non-starch polysaccharide (NSP) in wheat bran, accounting for approximately 70% of the NSP of wheat bran [5]. Ferulic acid (FA) is the main phenolic compound present in the bran and attached to arabinose in the AX side chain. In addition, FA has antioxidant, antithrombotic, and anticancer activities [6], which means that AX has a variety of nutritional and health effects, such as moistening the bowel and enabling defecation, reducing cholesterol, regulating blood sugar, anti-oxidation, and immune regulation [7]. According to its solubility in water, AX was classed into water-unextractable AX (WUAX) and water-extractable AX (WEAX).

Many researchers reported that the rough texture of whole wheat foods was due to the AX in wheat bran. Therefore, more and more attention was paid to improving the texture through physical or chemical methods. Reports showed that the viscoelasticity and the thermal aggregation of gluten was improved by WEAX during heating. Thus, it contributed to the dough network’s structural compatibility, among which it enlarged the loaf volume and made the textural property of Chinese steamed bread softer [8]. However, the WUAX, which accounted for 90% of total AX in the wheat bran, has significant adverse effects on the gluten network formation, especially weakening the gas holding capacity during heating. Therefore, it results in a rough texture, a lower specific volume, and an undesirable appearance of whole wheat-based products. Wang et al. found that WUAX competed with gluten for water to weaken the attraction between gluten protein molecules, which indirectly interfered with the formation of the gluten network [9]. Arif et al. also found that WUAX weakened the sensory and physical properties of the bread, affecting the quality of the final product [10]. In addition, our previous study reported that the WUAX could damage the viscoelasticity and the thermal properties of gluten, which was attributed to the competition with gluten for water and the disruption of the covalent cross-linking caused by WUAX, making whole wheat-based foods poor [11]. Moreover, Jiang et al. found that WUAX hydrolyzed by pentosanase was beneficial to the formation of uniform and fine crumb structures, leading to the higher volume and lower firmness of wholewheat Chinese steamed bread [12].

In summary, the mechanism of WUAX’s action in dough mainly focused on the interaction between WUAX and the main ingredients of flour, such as starch or gluten. However, the detailed mechanism of WUAX on the gluten components, especially on gliadins and glutenins, is still unclear and needs to be further explored regarding the quality deterioration mechanism in whole wheat-based foods. Therefore, in this work, the gluten protein was further separated into gliadins and glutenins. In addition, the interaction between WUAX and gluten protein components was observed by thermal analysis, secondary structure content, rheological behavior, and other methods in order to provide theoretical support for the subsequent improvement of whole wheat-based products.

## 2. Materials and Methods

### 2.1. Materials and Chemicals

Gluten (protein content was 70%) and wheat bran were gained from Yihai Kerry Co., Ltd. (Kunshan, China). 5,5′-dithiobis (2-nitrobenzoic acid) (DTNB) were purchased from Yuanye Co., Ltd. (Shanghai, China). The solvents and the remaining chemicals were of analytical quality.

### 2.2. Exaction of Water-Unextractable Arabinoxylans

The WUAX was obtained according to the method of Si et al. with some modifications [11]. First, wheat bran was defatted with 90% ethanol, and the step was repeated three times. Next, starch and protein in the wheat bran were removed with amylase and protease, followed by heat inactivation of the enzyme (100 °C, 30 min). The pretreated bran was then dried at 45 °C in a low-temperature oven. WUAX was extracted with saturated barium hydroxide at 1:15 (*w*/*w*), and the extracted suspension was adjusted to pH 5 with acetic acid in order to precipitate the protein. Subsequently, the supernatant was dialyzed (Mw cut-off 14 kDa) at 4 °C for 72 h to remove small molecular salts, and the crude polysaccharide solution was then concentrated using a vacuum rotary evaporator. It was precipitated with a final ethanol concentration of 65% (*v*/*v*) and freeze-dried with a vacuum freezing dryer. Finally, the solid was ground through an 80-mesh sieve, and the obtained AX powder was stored in a desiccator. The WUAX content was 82.01%, as measured by the orcinol-hydrochloric acid method, and its moisture content was 4.10 ± 0.49%.

### 2.3. Preparation of Wheat Gliadins and Glutenins

Glutenins and gliadins were separated from gluten according to the method of traditional Osborne-Mendel separation. Gluten was defatted with n-hexane, and the step was repeated three times. The defatted gluten (100 g) was added to 2400 mL of 65% ethanol, stirred at 35 °C for 3 h, and centrifuged at 5000× *g* for 20 min. The ethanol in the supernatant was then removed using a rotary evaporator at 45 °C to obtain the gliadins solution. Glutenins in lower precipitation were extracted with 1200 mL of deionized water using 0.1 M NaOH solution to adjust to pH 10. It was stirred at 50 °C for 3 h, and centrifuged at 5000× *g* for 20 min. The obtained supernatant was added to a final ethanol concentration of 65% (*v*/*v*), and glutenins were obtained by the precipitation method, which used 0.1 M HCl solution to adjust pH at 7. The glutenins were then washed with 400 mL of deionized water and stirred at room temperature for 2 h, followed by centrifugation at 5000× *g* for 20 min to discard the supernatant. The above step was repeated twice to obtain glutenins. The precipitations of gliadins and glutenins were freeze-dried, ground, and passed through a 60-mesh sieve. The moisture contents of gliadins and glutenins (5.97 ± 0.29% and 6.51 ± 0.44%, respectively) were measured by 44-15A (AACC, 2000). The crude protein contents of gliadins and glutenins (92.54 ± 0.64% and 90.73 ± 0.49% (wet basis, *w*/*w*), respectively) were measured by the Kjeldahl method (GB 5009.5-2016).

### 2.4. Sample Preparation

Gliadins (5.0 g) or glutenins (5.0 g) were separately mixed with different amounts of WUAX (0, 100, 200, 300, 400, and 500 g/kg (protein-based)). 15 mL of deionized water was added to the mixed powder and stirred well. The mixture was stirred at room temperature for 3 h and stored at 4 °C overnight. Subsequently, all samples were lyophilized with a vacuum dryer, ground, and passed through an 80-mesh sieve. The final sample (Gliadins-WUAX and Glutenins-WUAX abbreviated as Glia-W and Glu-W) concentrations of WUAX were 0, 2, 4, 6, 8, and 10% (*w*/*w*, wet weight of protein). The lyophilized powder was used to determine chemical interaction, the free sulfhydryl (free SH) content, thermal properties, intrinsic fluorescence spectra, and fourier transform infrared (FTIR) spectroscopy analysis. The sample without WUAX served as the control group (abbreviated as Glia and Glu). Samples with WUAX were abbreviated as glia-W and glu-W.

### 2.5. Determination of Free SH Content

Based on the method of Feng et al., the free SH content of all samples was determined [13]. The freeze-dried protein powder (30 mg) was added to 5.5 mL Ellman’s reagent [250 mmol/L Tris-HCl buffer (pH 8.5), propan-2-ol and 4 g/L DTNB) in ethanol (5/5/1, *v*/*v*/*v*)], stirred for 30 min at room temperature in the dark, and centrifuged at 4800× *g* for 10 min. Spectrophotometer (T9 type, Puchan universal instrument Co., Ltd., Beijing, China) was used to measure the absorbance value of supernatant at 412 nm. Finally, the free SH content was obtained by the following formula:$${\mathrm{SH}}_{\mathrm{free}}\;\mathrm{content}=\frac{A*V}{\varepsilon *b*m}$$
where *A* is the absorption of the sample at 412 nm; *m* is sample mass, g; *ε* is the molar absorption coefficient, *ε* = 13,600 M^−1^cm^−1^; *V* is the total volume of samples during determining free SH, L; and *b* is the thickness of the cuvette, *b* = 1 cm.

### 2.6. Measurement of Intrinsic Fluorescence Spectra

The intrinsic fluorescence spectra were determined, according to the method of Guo et al., with slight modifications [14]. WUAX solutions (2, 4, 6, 8, and 10 g/L) were prepared. Gliadins (500 mg) were dissolved in 650 mL/L ethanol (250 mL), and glutenins (500 mg) were dissolved in 0.1 M NaOH solution (250 mL) and centrifuged. WUAX solution (1 mL) was added to the supernatant (1 mL) and diluted to 10 mL. The protein solution without WUAX was served as the control group. The solution was set at room temperature for 10 min in the dark. Spectrofluorometer (F-7000, Hitachi, Ltd., Tokyo, Japan) was used to measure the intrinsic fluorescence spectra from 300 to 500 nm at room temperature. The emission was excited at 280 nm, and the slits of excitation and emission were set at 5 nm.

### 2.7. Determination of Gliadins and Glutenins Solubility in Different Solvents

Non-covalent interaction was determined according to Wang et al. with some modifications [15]. Different selective buffer dissolved proteins were used to disrupt ionic, hydrogen bonds or hydrophobic interactions, prepared in phosphate buffer (0.05 M, pH = 7.0) as follows: (PA) 0.05 M NaCl; (PB) 0.6 M NaCl; (PC) 0.6 M NaCl and 1.5 M urea; and (PD) 0.6 M NaCl and 8 M urea. Protein powder (200 mg) was added to each solvent (10 mL) and stirred for 1 h at 25 °C, followed by centrifugation at 10,000× *g* for 20 min. The bicinchoninic acid (BCA) protein assay kit was used to measure the protein concentration in the supernatant. A standard was prepared with bovine serum albumin, expressed as g soluble protein/L of supernatant. The difference between soluble gliadins or glutenins in PB and PA, PC and PB, and PD and PC represented ionic bonds, Hydrogen bonds, and Hydrophobic interactions.

### 2.8. Fourier Transform Infrared Spectroscopy

FTIR was used to research the secondary structure of the protein. 10.0 mg protein was mixed with 1.0 g KBr, ground, and then pressed into a slice. The FTIR spectrometer (Antaris Ⅱ, Thermo Nicolet Corporation, Madison, WI, USA) was used to obtain the spectrum over the wavelength range of 400 cm^−1^ to 4000 cm^−1^ with 32 scans and at 4 cm^−1^ resolution. Omnic software package (version 8.0) and PeakFit software (version 4.12) was used to analyze the secondary structures of all samples [16]. The amide I region spectra were classified into 1660–1700 cm^−1^, 1650–1659 cm^−1^, 1640–1650 cm^−1^, and 1610–1640 cm^−1^, corresponding to β-turns, α-helixes, random coil, and β-sheets, respectively [17].

### 2.9. Dynamic Rheological Measurements

According to the method reported by Wang et al., with slight modifications [15], the rheological properties of gliadins and glutenins were determined by the rotational rheometer (DHR-3, TA Instrument, New Castle, DE, USA). Dynamic rheological measurements were performed using freshly prepared samples of gliadins and glutenins with/without WUAX. First, the linear viscoelastic region of the protein was obtained by stress scanning over frequencies ranging from 0.1 to 100 Hz. Subsequently, the sample was placed between 40 mm steel plates (1 mm gap) and allowed to rest for 20 min to relax the residual stresses. The edge of all samples was covered with a thin layer of silicone oil in order to avert moisture loss during testing. The frequency sweep was measured over the range of frequency 0.1–100 Hz at 25 °C, within the linear viscoelastic region at a constant strain of 0.1%.

### 2.10. Thermal Properties Analysis

According to the method reported by Feng et al. [13], the thermal properties of glutenins and gliadins were analyzed by thermos-gravimetric analysis (TGA). A Mettler Toledo Star (Mettler Toledo Corp., Greifensee, Switzerland) was used to obtain TGA analysis for all samples (approximately 10 mg), among which a heating rate of nitrogen atmosphere was set from 50 to 600 °C at 10 °C /min. STAR software (version 9.01) was used to analyze the degradation temperature (T_*d*_) and weight loss rate.

### 2.11. Statistical Analysis

The results were described as a mean of three replicates ± SD (standard deviation) and evaluated for their statistically significant difference with ANOVA using SPSS 26 software, where *p* < 0.05 represented statistical significance. All figures were obtained by origin 2018 software.

## 3. Results and Discussions

### 3.1. Free SH Content Analysis

The content of free SH groups can be used to characterize the degree of protein aggregation, and its content is negatively correlated with the degree of gluten protein aggregation [18]. Extra WUAX has obvious modification effects on the free SH contents of gliadins and glutenins, which are summarized in Figure 1. With 10% of WUAX, the SH content of gliadins increased by 25.5% compared with the control (0.98 μmol/g), which demonstrated that WUAX inhibited the formation of SS bonds and loosened the molecular structure of gliadins. The free SH content in glutenins remarkably increased by 65.2% compared with the control group (1.15 μmol/g) in accordance with the variety of gliadins. This trend showed that WUAX prompted the unfolding or denaturation of the structure of glutenins. The addition of WUAX had a more significant effect on the content of free SH in glutenins than that of gliadins. This was due to the intermolecular SS bonds of glutenins being readily damaged by the external environment [19]. Our results of the free SH content for gliadins and glutenins with and without WUAX were consistent with those of Guo et al., whose results suggested that additional inulin restrained the formation of the SS bonds in glutenins and gliadins and loosened the molecular structure of glutenins and gliadins [14]. Therefore, the enhancive free SH content of the gluten protein components caused by WUAX may be related to the large steric hindrance of WUAX. The physical entanglement between WUAX and protein decreased the chance of free SH groups contacting each other and weakened binding interactions between proteins, which hindered the formation of SS bonds [11].

### 3.2. Intrinsic Fluorescence Spectra Analysis

Intrinsic fluorescence spectrum analysis can be used to characterize the microenvironment changes of a fluorescent amino acid, and it serves as an essential indicator to characterize proteins based on their intermolecular interactions, dynamics and conformation [15].

The effect of WUAX on the intrinsic fluorescence intensity (*I_max_*) and maximum fluorescence absorption wavelength (*λ_max_*) of gliadins and glutenins is shown in Table 1. Compared with the gliadins group, the *λ_max_* of gliadins with WUAX changed slightly, and there was no significant difference. These indicated that WUAX had a weak effect on the microenvironment of tryptophan residues in gliadins, but the addition of WUAX affected the hydration process of gliadins, making the structure of gliadins unfold and loosen. However, the *I_max_* of gliadins with WUAX increased, which means that WUAX increased the exposure degree of fluorescent chromophobe groups in gliadins. In particular, the increase in AX content further hindered the aggregation between the subunits of gliadins and aggravated the structural changes of the gliadins.

For glutenins, when WUAX content was less than <60 g/kg, the *λ_max_* of glutenins experienced no significant changes. However, the *λ_max_* of glutenins occurred the red shift with the high level of WUAX. This was likely because that the increased concentration of WUAX caused that WUAX to come into contact with amino acids in the hydrophobic core of glutenins, and its hydroxyl group on the side chain increased the polarity of the tryptophan residue microenvironment. After the addition of WUAX, the *I_max_* of glutenins with WUAX was the opposite of gliadins. A regular decreasing trend was observed (Table 1) with the increase in WUAX level. However, the change of free SH (Figure 1) showed that WUAX obstructed cross-linking of glutenins, resulting in protein structure unfolding. Therefore, we speculated that WUAX caused the decrease in the *I_max_* of glutenins, probably because WUAX interacted with glutenins to form a new complex, which led to an obvious quenching effect of tryptophan groups on the surface of glutenins.

Si et al. reported that the addition of WUAX caused a decrease in the *I_max_* and a red shift of gluten protein [11]. As a result, we speculated that the hydrophilicity of WUAX’s side chain hydroxyl group altered the polarity of the environment around tryptophan residues in gliadins and glutenins. However, due to the network structure of glutenins, a more specific surface area came into contact with WUAX, which resulted in the decrease in gluten fluorescence intensity. Therefore, we guessed that the addition of WUAX changed the conformation, dynamics, intermolecular interactions, and the microenvironment of glutenins and gliadins, thereby resulting in the gluten protein structure change.

### 3.3. Non-Covalent Interaction Analysis

Non-covalent interactions between proteins are critical in maintaining the three-dimensional structure and stability of the protein complex. Since different concentrations of solvents (NaCl and urea) have a destructive effect on the ionic bonds, hydrogen bonds, and hydrophobic interactions in protein intramolecular or intermolecular, the solubility of the protein in different solvents indirectly expresses the stabilizing force of the protein (hydrogen bonds, hydrophobic interactions, and ionic bonds).

The effect of WUAX on the non-covalent bond between wheat glutenins and gliadins is shown in Figure 2. Hydrophobic interactions played an critical role in maintaining the conformation of gliadins [20]. However, the main non-covalent forces in glutenins were hydrogen bonds and hydrophobic interactions (Figure 2B). The phenomenon suggested that it was pivotal for maintaining the conformation of gliadins and glutenins [21], resulting from gliadins and glutenins containing abundant non-polar amino acids, and leading to more hydrophobic interactions at hydration [22]. The ionic bond strength of the two proteins was relatively low compared with hydrophobic interactions and hydrogen bonds (Figure 2), which indicated the presence of weak ionic bonds in proteins formed by few ionizable amino acid residues [23]. Due to the large molecular weight of glutenins, the concentration of glutenins dissolved in different solvents was lower than that of gliadins.

After adding WUAX to gliadins, it was observed that the ionic bond strength of gliadins decreased, and 4% WUAX had the lowest strength. The hydrogen bonds between proteins can be broken into low urea concentrations, increasing protein solubility. As the urea concentration increased, the non-covalent interactions were further destroyed, resulting in protein solvation [11]. As the amount of WUAX increased, weaker hydrogen bonds were formed, but the hydrophobic interaction decreased, except for the 10% WUAX. This phenomenon may result from the large steric hindrance of WUAX, which presents a linear structure in an aqueous solution. They can entangle and hinder mutual contact between gliadins, resulting in the reduction of hydrophobic interactions, and this facilitates the formation of hydrogen bonds within gliadin molecules. A study reported by Guo et al. found that inulin with a high molecular weight could hinder the molecular movement of gluten protein and facilitate the formation of weaker hydrogen bonds [14]. However, the high steric hindrance inhibited the formation of ionic bonds and hydrophobic interactions in gluten protein and prevented the aggregation of gluten protein.

After adding WUAX to glutenins (Figure 2B), it was observed that the hydrogen bonds had decreased. However, the hydrophobic interaction increased and then decreased in accordance with the changing trend of the ionic bond of glutenins. Nonetheless, the strength of the ionic bond and the hydrophobic interaction were higher than the control, except for the 10% WUAX. However, the changing trend of glutenins was opposite to that of gliadins. This may be due to the structural differences between glutenins and gliadins. Since the network structure of glutenins was more likely to be broken by WUAX, the addition of WUAX changed the microenvironment of the hydrophobic amino acids inside the protein, so the protein structure would be unfolded. However, due to the larger steric hindrance of WUAX, the higher addition of WUAX entangled to form a physical barrier on the surface of the glutenin molecules, hindering their structure extension and the form of hydrogen bonding between the glutenins. We thus concluded that WUAX mainly destroyed the hydrophobic interactions of gliadins and the hydrogen bonds of glutenins. Si et al. reported that the effect of adding WUAX on the non-covalent force of gluten protein, which is consistent with the results of glutenins; this indicated that WUAX might affect the network structure of gluten protein mainly by changing the spatial conformation of glutenins, leading to the poor quality of the final product [11].

### 3.4. Secondary Structure of Protein Analysis

FTIR can quickly analyze the secondary structure content of protein in the amide I band (1600–1700 cm^−1^). The characteristic absorption peaks of the amide I band are shown in Figure 3, and the secondary structure content is summarized in Table 2. It was observed that the protein’s secondary structure in all samples was mainly β-sheets and β-turns structure, and random coils and α-helixes account for a low fraction, which was consistent with the results of Feng et al. [13].

Table 2 illustrates that there was a fluctuation within a narrow range in the random coil structure of gliadins after adding WUAX. The β-sheets content and α-helixes decreased, and the β-turns content increased subjected to the WUAX treatment, which indicated that the increasing β-sheets were at the expense of β-turns after the addition of WUAX. Ang et al. reported that β-turns could change the orientation of protein-peptide chains and achieve multiple reversals of protein structure, thereby making gliadins appear as prolate ellipsoidal protein [24]. Guo et al. reported that the lower β-turns in gliadins caused by adding KGM was helpful to the stability of gliadins compared with the control group [14]. In addition, according to the fact that the structure of the β-sheets was the most stable structure [25], our results demonstrated that adding WUAX decreased the stability of gliadins through the change of secondary structure. Interestingly, when the additional amount of WUAX was more than 4%, β-sheets and β-turns contents had little change. This may be due to the mutual entanglement of WUAX to form a larger physical barrier on the surface of gliadins, which also prevented WUAX from destroying the structure of the gliadins, so increasing the amount of WUAX (more than 6%) would not damage the secondary structure of gliadins.

Glutenins presented a fiber structure; thus, the structure of β-sheets played a critical role in maintaining the secondary structure of glutenins [26]. As demonstrated in Table 2, the content of the β-sheets presented a decreased trend; the random coils and α-helixes had a slight increase and decrease, respectively, compared with the control. The changing trend of β-turns was the opposite of β-sheets. Interestingly, changes in all structures were relatively low when the addition of WUAX was more than 6%. This confirmed that the spatial conformation of the conformation change of glutenins caused by WUAX was limited, which accorded with the change of gliadins. Moreover, Liu et al. studied the effects of inulin addition with different degrees of polymerization on the contents of the secondary structure of gliadins and glutenins [27]. With the increase in the degree of inulin polymerization, the β-sheets of gliadins and glutenins increased, while the β-turns trend was the opposite.

Based on the changes of free SH, we thought that WUAX induced protein structural rearrangement at lower levels, reducing the chance of free SH contact between glutenins. With the increase in WUAX content, the changes in the secondary structure of proteins tended to be stable, suggesting that the increase in free SH content was likely because the physical entanglement of WUAX prevented the cross-linking of glutenins.

### 3.5. Dynamic Rheological Analysis

Gluten protein is composed of glutenins and gliadins, which is a critical component of wheat protein. As we all know, gliadins confer viscosity and extensibility properties, while glutenins impart elasticity and strength properties [28]. Therefore, the dynamic rheological measurement could be used to evaluate the effects of WUAX on the viscoelasticity of the two proteins. Figure 4 provides the elastic modulus (storage modulus G′) and viscous modulus (loss modulus G″) as a function of the frequency for all samples at 25 °C subjected to the WUAX treatment.

As shown in Figure 4, after the addition of WUAX, the viscoelasticity of gliadins and glutenins with WUAX was lower than the control group. However, it was reduced and then increased with the increase in the WUAX. When the WUAX was 4% and 6%, respectively, it reached the lowest point. Unlike the glutenins, the effects of the WUAX amounts on the viscoelasticity of gliadins were not dramatically distinct. In addition, after the addition of WUAX, the tan δ of gliadins was increased, followed by the decrement, but it was higher compared with the control group (Figure 4C), which was consistent with the change of viscosity and elasticity. Therefore, WUAX damaged the elasticity of gliadins more readily and formed the soft and viscous gliadins [8], making the rheological properties of gliadins more liquid-like [20]. The reason may be that WUAX, a hydrophilic macromolecule polysaccharide, had a higher viscosity, which can maintain the viscosity of gliadins. For glutenins (Figure 4F), the tan δ gradually increased with increasing WUAX, which indicated that G′ was close to and just above G″. Thus, the samples with WUAX displayed the typical weak gel properties, which suggested that the weak junction zone can easily be damaged at a low shear rate [29]. The addition of WUAX had a more significant effect on the viscosity of glutenins than on the elasticity from Figure 4D,E. This might be due to the high viscosity of WUAX, which can bring some compensation to the viscosity damage of the protein. In addition, compared with gliadins, the relative solid-like behavior of glutenins via the tan δ analysis, was caused by the intermolecular disulfide bonds in glutenins, which created a strong structure for glutenins [30].

Compared with gliadins, the WUAX amount obviously influences glutenins, which might be related to the structure of glutenins and gliadins. There were more sites to form non-covalent interactions with WUAX in that glutenins had a larger molecular weight and a certain network structure. The viscoelasticity of both proteins obviously decreased, which is in harmony with the result of free SH content (Figure 1). Moreover, WUAX could interact with gluten via non-covalent bonds (hydrogen bonds and hydrophobic interaction), inhibiting the cross-linking of protein structure, thereby reducing their ductility. On the one hand, it was related to the good water holding capacity of WUAX. WUAX competed with protein for water, limiting the hydration of protein, and it finally resulted in the softening of the protein (lower G′ and G″) [31]. On the other hand, longer chains of WUAX made it difficult to establish a cross-link between gliadins/glutenins and polysaccharide polymers. Therefore, it could be used to explain that whole grains would result in a lower specific volume of whole wheat-based foods during the heating process.

### 3.6. Thermal Properties Analysis

The effects of WUAX additions on the thermal properties of gliadins and glutenins were investigated by TGA analysis. The weight loss profiles of various proteins are shown in Figure 5. The weight loss at 600 °C and the degradation temperature (T*_d_*) obtained from the TGA profile were the main parameters for the characterization of the protein thermal properties, as shown in Table 3. The weight loss of gliadins and glutenins exhibited the condition of the protein structure. The higher weight loss presented the protein network structure as more open and weaker, while a decrease in the weight loss can be related to the formation of a stronger and more compact protein structure [32]. Our results showed that protein weight loss was first increased and then decreased after adding WUAX, but that it was higher than that of the control (except for the gliadins with 10% WUAX). Feng et al. reported that extra wheat bran increased the weight loss of gliadins, glutenins, and GMP, which was consistent with our result [13]. Therefore, the lower amount of WUAX made the structures of the gliadins and glutenins more open and weaker, which might be because the WUAX hindered the covalent interaction (SS bonds), leading to the stability of the two proteins declining. With the increasing WUAX, it can become entangled between proteins, resulting in structure stability recovery, according to the dynamic rheological result (Figure 4A,B,D,E).

The TGA profiles of glutenins and gliadins can be divided into two steps (Figure 5): the water evaporation stage at 50–150 °C and the cleavage stage of peptide bonds, disulfide bonds, O-O, and O-N bonds in the protein at 150–600 °C [32]. The T*_d_* of samples from TGA is summarized in Table 3. The result indicated that T*_d_* was decreased and then increased with the addition of WUAX compared with the control group, which was not significant, and which was consistent with the result of Zhao et al. [8]. They reported that both the weight loss rate and the T*_d_* of gluten were not significantly changed when subjected to the WEAX treatment. Similarly, Si et al. also reported that the T*_d_* of gluten with WUAX exhibited a slight variation compared with the control group and did not form new compounds, suggesting that WUAX kept the thermal stability of gluten molecules [11]. Interestingly, our result differed from that of Guo et al., who found that the T*_d_* of wheat gliadins increased after adding inulin, while that of glutenins decreased slightly, which may be related to the changes of disulfide bonds in the protein caused by inulin [33]. Therefore, adding WUAX results in the structures of gliadins and glutenins being more open, but the effect on the stability of the proteins was not significant, which might be related to the non-covalent interaction between gliadins/glutenins and WUAX.

## 4. Conclusions

The study showed that the free SH content of glutenins and gliadins increased after the addition of WUAX, suggesting that WUAX hindered covalent interaction between proteins resulting in a more fragile and open network structure. However, the changing trend of non-covalent interaction (mainly hydrophobic interactions and hydrogen bonds) between glutenins and gliadins was the opposite when subjected to the WUAX treatment. WUAX might promote the formation of more weakened hydrogen bonds for gliadins, while it adversely influences the formation of hydrogen bonds in glutenins. The rheological results showed that the addition of the WUAX damaged the viscoelasticity of proteins, and the effect of the amount of WUAX on the viscoelastic behavior of the gliadins was lower than that of glutenins, which may be due to the structural differences between the two proteins. Interestingly, the tan δ of the glutenins increased with the increasing amount of WUAX, which was related to the higher viscosity of the WUAX itself. When the WUAX acted alone with glutenins, it partially compensated for the loose structure of the glutenins due to the absence of gliadins as the binding agent of the glutenins. Moreover, the reduction of β-sheets content for gliadins and glutenins weakened the rigid structure. Thermal property analysis suggests that the structure became more open, and the stability of the two proteins decreased after the addition of WUAX. Therefore, we speculated that after the addition of WUAX, a small amount of WUAX has a great impact on the covalent interaction of protein. With increasing WUAX, non-covalent interactions (hydrogen bond and ionic bond) between WUAX and protein played a critical role in the structure of the two proteins. The interactions between WUAX and independent gluten components are a matter of great significance for research into and development of WUAX-gluten gel products; they also provide a theoretical reserve for enhancing the quality of whole wheat-based foods. The effect of WUAX on the nutritional and digestive properties of gliadin and glutenin gels requires further investigation.

## Figures and Tables

**Figure 1 foods-12-01800-f001:**
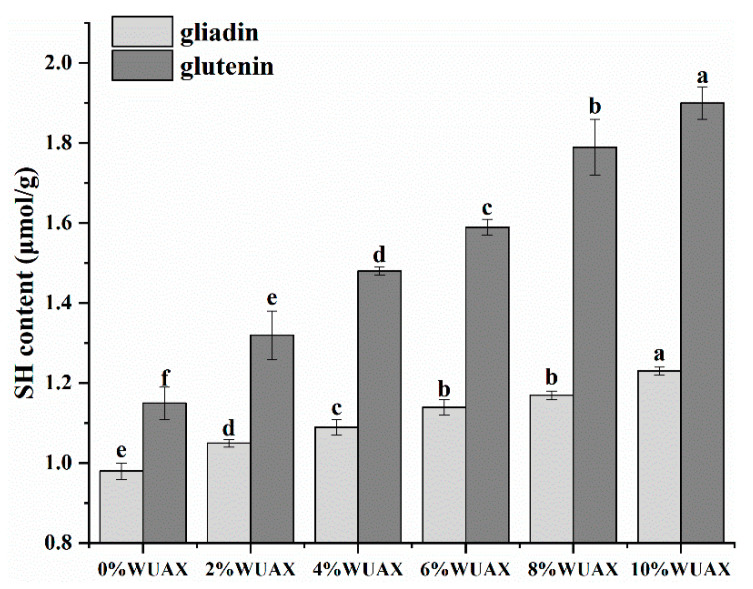
The effect of water un-extractable arabinoxylan (WUAX) on free sulfhydryl (free SH) content of gliadin and glutenin. Data are means of three independent experiments (*n* = 3) ± SD. Different letters above the bar mean significant differences (*p* < 0.05).

**Figure 2 foods-12-01800-f002:**
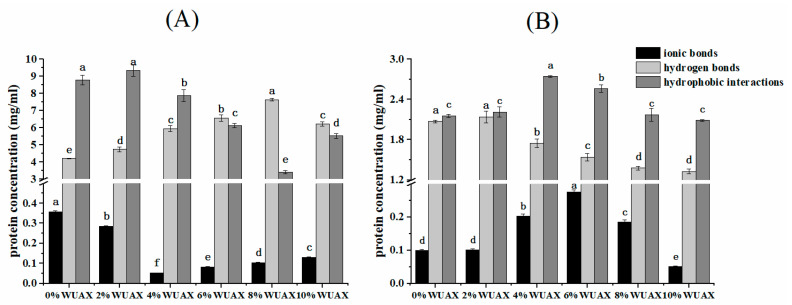
The effect of a different amount of WUAX on the non-covalent interaction of gliadin (**A**) and glutenin (**B**). Note: Data are means of three independent experiments (*n* = 3) ± SD. Different letters above the bar mean significant differences (*p* < 0.05).

**Figure 3 foods-12-01800-f003:**
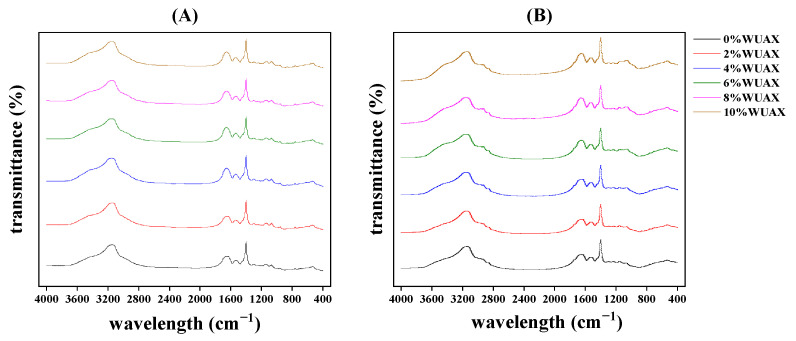
FTIR profiles of gliadin (**A**) and glutenin (**B**) treated with different concentrations of WUAX.

**Figure 4 foods-12-01800-f004:**
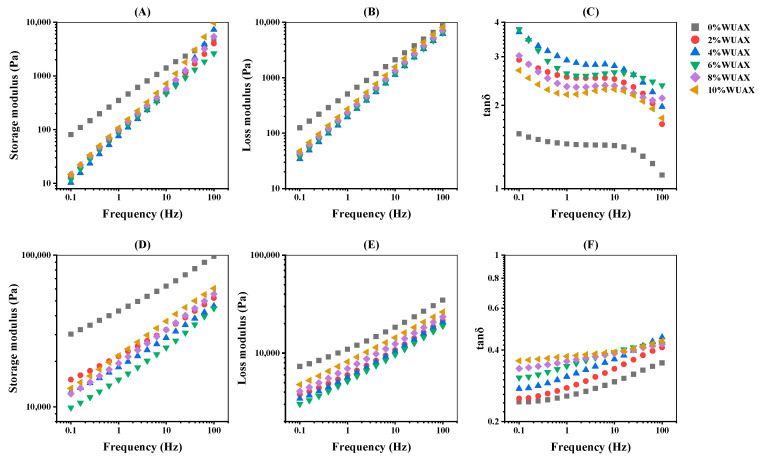
Frequency sweeps of G′ (**A**,**D**), G″ (**B**,**E**), and tan δ (**C**,**F**) of gliadin (**A**–**C**) and glutenin (**D**–**F**) with different amounts of WUAX. Note: Data are the mean of three independent experiments (*n* = 3).

**Figure 5 foods-12-01800-f005:**
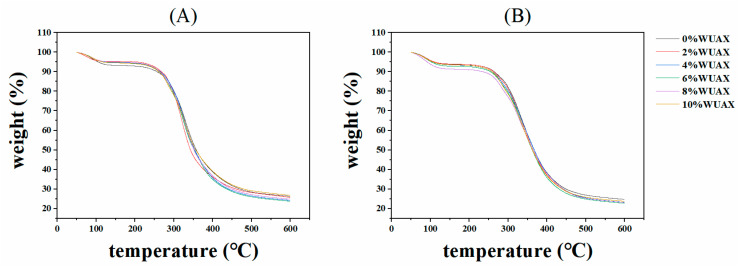
Weight loss profiles of gliadin (**A**) and glutenin (**B**) with different amounts of WUAX.

**Table 1 foods-12-01800-t001:** Fluorescence spectra profiles (*I_max_* and *λ_max_*) of gliadin and glutenin.

Sample	WUAX Content	*λ_max_*	*I_max_*
Glia-W	0%	339.5 ± 0.3 ^a^	1941.0 ± 2.6 ^a^
	2%	339.6 ± 1.0 ^a^	1969.7 ± 8.0 ^b^
	4%	339.7 ± 0.4 ^a^	1994.7 ± 6.4 ^c^
	6%	340.1 ± 0.9 ^a^	2011.7 ± 6.7 ^d^
	8%	340.2 ± 0.5 ^a^	2069.7 ± 8.0 ^e^
	10%	340.5 ± 0.8 ^a^	2091.7 ± 12.7 ^f^
Glu-W	0%	339.8 ± 1.0 ^b^	2190.7 ± 6.4 ^a^
	2%	339.3 ± 0.7 ^b^	2162.0 ± 15.0 ^b^
	4%	339.8 ± 1.4 ^b^	2106.3 ± 17.2 ^c^
	6%	340.3 ± 1.3 ^b^	2024.3 ± 13.5 ^d^
	8%	343.7 ± 0.5 ^a^	1964.7 ± 12.4 ^e^
	10%	343.7 ± 1.9 ^a^	1936.3 ± 15.0 ^f^

Data are mean ± SD (*n* = 3). Different lowercase letters in the same column mean significant differences (*p* < 0.05). Glia-W means gliadin-WUAX and Glu-W means glutenin-WUAX (the same below).

**Table 2 foods-12-01800-t002:** Secondary structure of gliadin and glutenin with different amounts of WUAX.

Sample	WUAX Content	β-Sheets	α-Helixes	Random Coils	β-Turns
Glia-W	0%	43.27 ± 0.42 ^a^	13.80 ± 0.28 ^a^	13.28 ± 0.25 ^a^	29.67 ± 0.94 ^c^
	2%	42.35 ± 0.48 ^a^	12.50 ± 0.32 ^b^	12.31 ± 0.01 ^c^	32.85 ± 0.79 ^b^
	4%	38.93 ± 0.20 ^b^	12.68 ± 0.09 ^b^	12.97 ± 0.08 ^b^	35.43 ± 0.37 ^a^
	6%	37.19 ± 0.10 ^c^	12.55 ± 0.57 ^b^	13.30 ± 0.04 ^a^	36.97 ± 0.63 ^a^
	8%	37.49 ± 0.06 ^c^	12.18 ± 0.09 ^b^	13.36 ± 0.08 ^a^	36.98 ± 0.04 ^a^
	10%	37.64 ± 0.73 ^c^	11.98 ± 0.23 ^b^	13.37 ± 0.06 ^a^	37.02 ± 0.56 ^a^
Glu-W	0%	47.42 ± 0.29 ^a^	12.32 ± 0.20 ^a^	12.09 ± 0.57 ^c^	28.17 ± 0.09 ^c^
	2%	46.25 ± 0.39 ^b^	12.24 ± 0.31 ^ab^	12.17 ± 0.38 ^c^	29.33 ± 0.33 ^bc^
	4%	46.09 ± 0.14 ^b^	11.85 ± 0.20 ^bc^	12.35 ± 0.03 ^bc^	29.71 ± 0.02 ^b^
	6%	43.77 ± 0.51 ^c^	11.49 ± 0.09 ^c^	12.85 ± 0.42 ^abc^	31.89 ± 1.02 ^a^
	8%	42.88 ± 0.02 ^d^	11.40 ± 0.08 ^c^	13.34 ± 0.08 ^ab^	32.38 ± 0.03 ^a^
	10%	42.67 ± 0.25 ^d^	11.51 ± 0.03 ^c^	13.58 ± 0.59 ^a^	32.24 ± 0.87 ^a^

Data are mean ± SD (*n* = 3). Different lowercase letters in the same column mean significant differences (*p* < 0.05).

**Table 3 foods-12-01800-t003:** TGA profiles [weight loss at 600 °C and degradation temperature (T*_d_*)].

Sample	WUAX Content (g/kg)	Weight Loss (%)	T*_d_* (°C)
Glia-W	0%	73.74 ± 0.49 ^c^	329.7 ± 1.13 ^a^
	2%	74.34 ± 0.40 ^bc^	328.7 ± 1.56 ^ab^
	4%	75.78 ± 0.38 ^a^	328.8 ± 2.12 ^ab^
	6%	76.23 ± 0.34 ^a^	324.8 ± 2.41 ^b^
	8%	75.15 ± 0.17 ^ab^	327 ± 0.85 ^ab^
	10%	73.23 ± 0.27 ^c^	331.2 ± 1.83 ^a^
Glu-W	0%	75.28 ± 0.36 ^b^	333.2 ± 0.71 ^a^
	2%	76.80 ± 0.27 ^a^	331.3 ± 1.41 ^ab^
	4%	76.79 ± 0.21 ^a^	328.7 ± 0.99 ^b^
	6%	77.24 ± 0.16 ^a^	333.2 ± 1.13 ^a^
	8%	76.86 ± 0.38 ^a^	334.7 ± 2.26 ^a^
	10%	76.21 ± 0.23 ^a^	334.8±1.69 ^a^

Data are mean ± SD (*n* = 3). Different lowercase letters in the same column mean significant differences (*p* < 0.05).

## Data Availability

The datasets generated for this study are available on request to the corresponding author.

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
