# Peer review of "Entanglement between Water Un-Extractable Arabinoxylan and Gliadin or Glutenins Induced a More Fragile and Soft Gluten Network Structure"

_foods, 2023, doi:10.3390/foods12091800_

Round 1
Reviewer 1 Report
Dear authors
The study is good and carries an interesting idea, although it needs to diversify the sources and expand the number of repetitions in order to be generalized and comprehensive, in addition to that there are some observations that can be taken into account:
- - The study was conducted on the main components of gluten, gliadin and glutenin, and did not address starch, although the three components represent the basic polymer of flour.
- - The weight loss profiles of gliadin (A) and glutenin (B) were studied when treated with different amounts of WUAX under different temperatures degree. Other factors that may have an effective role such as moisture content, pH and/ or reaction time were omitted.
- - FTIR profiles of gliadin (a) and glutenin (b) were treated with different concentrations of WUAX, and the study did not expose proteolytic enzymes, as well as starch-degrading enzymes such as alpha and beta-amylase.
- - The headings in the Material and Methods pane do not match the Results pane.
With best regards
Author Response
1. First of all, thank you for your advices In this paper, the quality deterioration of whole wheat food is studied deeply. Based on previous studies, it was found that the degradation of gluten protein caused by AX was mainly due to the specific volume reduction and texture hardening caused by the collapse of gluten network structure. However, AX had a positive effect on the gelatinizing properties of starch. Therefore, this work focused on the effect of AX on gluten protein components.
2. Thank you for your valuable advice. The pH and reaction time of all samples were strictly the same, so they were not analyzed as variables in the study. For water content, referring to the analytical method of Wang et al. [1], it was found that water content was not explained in TGA. We accept their analytical method and believe that the overall weight loss rate of the protein is sufficient to reflect the unfolding and folding of the protein structure.
[1] Wang, P., Xu, L., Nikoo, M., et al. Effect of frozen storage on the conformational, thermal and microscopic properties of gluten: Comparative studies on gluten-, glutenin- and gliadin-rich fractions. Food Hydrocolloids, 2014, 35, 238-246.
3. Thank you for your valuable advice. In this work, gluten protein is a by-product of wheat starch processing. When wheat flour was classified and extracted by the three-phase horizontal snail process, gluten protein only contained a very small amount of α/β amylase due to the low molecular weight of the enzymes and the easy solubility of α/β amylase in water. Therefore, the influence of α/β amylase was ignored in the FTIR analysis of proteins in this experiment.
4. We agree with your suggestion and adjust the order of sections in the Material Methods. Thank you again for your review and suggestions on our articles.

Reviewer 2 Report
Dear Authors,
I would like to express my appreciation for your paper titled "Entanglement between water un-extractable arabinoxylan and gliadin or glutenins induced a more fragile and soft gluten network structure" which I had the pleasure of reviewing. I found the research aim, methods, and results presentation to be outstanding.
Your research provides an intriguing insight into the interaction between water un-extractable arabinoxylan and gliadin or glutenins, and its impact on the gluten network structure. The results of your study are highly relevant, especially considering the lack of research in this area.
I was impressed with the clarity of your methods section, which allowed me to easily understand the steps taken to reach the presented results. Your data presentation was also exemplary, providing a clear and concise representation of the study outcomes.
In summary, I would like to commend you on a well-executed and highly informative research paper.
Author Response
Dear Reviewer,
I am very glad that you have highly recognized the content of our work. Thank you very much for your careful review of the manuscript.
Best wish!

Reviewer 3 Report
The topic of the manuscript is very interesting, but I have a few comments.
Why the authors, when describing pentosans, do not mention anything about phenolic acid - ferulic acid. Some side chains of arabinose are esterified with phenolic acids, especially ferulic acid (FA). The presence of ferulic acid gives antioxidant properties. Occurrence of pentosans in wheat grain: endosperm, cell wall, and fruit and seed coat of wheat grains.
It should be remembered that the pro-health properties of pentosans in part depend on the connection with ferulic acid. This is missing in the introduction.
The methodology is well described, but in some cases the devices lack type, manufacturer, city and country.
In the results should be taken into account the ferulic acid arabinxylan complex.
Ferulic acid can also be determined in these preparations by HPLC.
In some cases there are too few references in the text.
There are still no SEM pictures to study the visual structure of the network.
Conclusions - correct
Author Response
1. The composition of AX was studied in the earlier stage, and the content of FA in AX extracted by Ba(OH)2 was lower than the detection limit. This is consistent with Xiao et al. [1], who reported that alkali extraction would lead to de-esterification of AX, resulting in a lower proportion of FA in the obtained AX. Therefore, in the Result and Discussion, FA was not introduced to explore the changes in protein structure.
This study is based on the existence of the whole wheat food in the process of processing the problem of deterioration. Therefore, the effect of AX extracted from wheat bran on protein structure was analyzed.
[1] Xiao X. Y., Qiao J. L., Wang J., et al., Grafted ferulic acid dose-dependently enhanced the apparent viscosity and antioxidant activities of arabinoxylan. Food Hydrocolloids, 2022, 128, 107557.
2. We appreciate your suggestion that FA has antioxidant properties in whole grain foods. Therefore, the introduction of FA in AX is supplemented in the introduction, which is shown in red font.
3. Thank you for your careful review of our manuscript. The instruments used in the Material and Methods have been supplemented in detail and are shown in red.
4. As the content of FA in AX used in this experiment is very low, it is not analyzed too much in the Results and Discussion.
5. Thank you for your suggestion. In the early stage of the experiment, HPLC was used to quantify FA in AX, but the results showed that the content of FA was lower than the detection limit.
6. Due to the lack of relevant studies, some references may be less. However, we agree with your suggestion very much. Now we should try our best to quote more references to increase the persuasive content. The supplementaries are marked in red.
7. Thank you for your advice. However, since the observation of the network structure is used in another research paper, it is a pity that the SEM cannot be used to characterize the protein network structure again in the design of this paper.

Round 2
Reviewer 3 Report
None.
Only, Please check the english language